# Glutamatergic Receptor Trafficking and Delivery: Role of the Exocyst Complex

**DOI:** 10.3390/cells9112402

**Published:** 2020-11-03

**Authors:** Matías Lira, Rodrigo G. Mira, Francisco J. Carvajal, Pedro Zamorano, Nibaldo C. Inestrosa, Waldo Cerpa

**Affiliations:** 1Laboratorio de Función y Patología Neuronal; Departamento de Biología Celular y Molecular, Facultad de Ciencias Biológicas, Pontificia Universidad Católica de Chile, 8331150 Santiago, Chile; mslira1@uc.cl (M.L.); rnmira@uc.cl (R.G.M.); francisco.carvajalc@outlook.cl (F.J.C.); 2Centro de Excelencia en Biomedicina de Magallanes (CEBIMA); Universidad de Magallanes, 01855 Punta Arenas, Chile; ninestrosa@bio.puc.cl; 3Laboratorio de Neurobiología, Departamento Biomédico, Facultad de Ciencias de la Salud, Universidad de Antofagasta, 1240000 Antofagasta, Chile; pedro.zamorano@uantof.cl; 4Instituto Antofagasta, Universidad de Antofagasta, 1240000 Antofagasta, Chile; 5Centro de Envejecimiento y Regeneración (CARE); Departamento de Biología Celular y Molecular, Facultad de Ciencias Biológicas, Pontificia Universidad Católica de Chile, 8331150 Santiago, Chile

**Keywords:** membrane trafficking, synapse, glutamatergic receptor, exocyst, constitutive trafficking, LTP-induced trafficking

## Abstract

Cells comprise several intracellular membrane compartments that allow them to function properly. One of these functions is cargo movement, typically proteins and membranes within cells. These cargoes ride microtubules through vesicles from Golgi and recycling endosomes to the plasma membrane in order to be delivered and exocytosed. In neurons, synaptic functions employ this cargo trafficking to maintain inter-neuronal communication optimally. One of the complexes that oversee vesicle trafficking and tethering is the exocyst. The exocyst is a protein complex containing eight subunits first identified in yeast and then characterized in multicellular organisms. This complex is related to several cellular processes, including cellular growth, division, migration, and morphogenesis, among others. It has been associated with glutamatergic receptor trafficking and tethering into the synapse, providing the molecular machinery to deliver receptor-containing vesicles into the plasma membrane in a constitutive manner. In this review, we discuss the evidence so far published regarding receptor trafficking and the exocyst complex in both basal and stimulated levels, comparing constitutive trafficking and long-term potentiation-related trafficking.

## 1. Introduction

In the central nervous system (CNS), the synapses are the specialized structures that connect neurons and let the flux of information between them. The main excitatory neurotransmitter in the brain is glutamate [1], and glutamatergic synapses comprise a presynaptic terminal, a specialized postsynaptic structures called dendritic spines [2], and glial cells (Figure 1). The presynaptic terminal contains synaptic vesicles where glutamate is stored and from here, glutamate is released to the synaptic cleft upon electrical stimulation [3]. The postsynaptic compartment contains the glutamate receptors, which are transmembrane proteins responsible for transducing the glutamate signal from extracellular space. 

There are a variety of glutamate receptors, both ionotropic and metabotropic [4], that are widely distributed in the brain and spinal cord in both neurons and glia [5,6,7,8,9,10,11]. Excitatory ionotropic neurotransmission is primarily mediated by α-amino-3-hydroxy-5-methyl-4-isoxazole propionic acid (AMPA) receptors and N-methyl-D-aspartic acid (NMDA) receptors. AMPA receptors are permeable to calcium, sodium, and potassium, with differential permeability depending on subunits compositions. Similar heterogeneity applies to NMDARs. They are mainly calcium-permeable, but similar to AMPARs, depending on the subunits’ composition, their properties change, giving to this channel unique properties that make it critical in brain development, synaptic plasticity, and neuropathology. AMPARs appears to have four agonist sites. Following the binding of the agonist to these binding sites, conformational changes occur, the channel opens, cation enters, and conductance increases, all within milliseconds [12,13]. Once open, the channel suffers rapid desensitization due to small conformational rearrangement in the dimer interface, which closes the pore and diminishes the conductance [13,14]. NMDARs are more restrictive in gating processes. NMDARs contain four agonist sites for glutamate and for co-agonist glycine [15]. If glycine is not present, the NMDAR opening does not occur. In addition, D-serine has been suggested to be a substitute for glycine in physiological NMDARs activation, mainly by activating synaptic NMDARs [16]. Additionally, the NMDAR channel is blocked by Mg^+2^ at physiological conditions. NMDAR opening relies on both agonist binding to their specific site and the release of Mg^+2^ due to AMPAR-mediated depolarization of the postsynaptic membrane [17]. In doing so, channels allow cation influx and evoke intracellular signaling. NMDARs have slow activation and desensitization compared to AMPARs, which has been related to synaptic memory after low-frequency synaptic inputs [18].

The trafficking and secretory mechanisms of neurotransmitter vesicles are well studied, but the constitutive and homeostatic trafficking and secretory mechanisms at or near the pre-synapse are much less understood [19]. On the other hand, the secretory pathway in the postsynaptic compartment has been far less described. The main challenge for the secretory pathway in the dendritic compartment is the distance from the soma and degree of arborization [20,21]. In the last two decades, we have more information about this secretory pathway, including mediators, organelle specializations, and protein machinery. The last step of the secretory pathway, i.e., the transition between cytoplasmic compartments and plasma membrane has received particular interest due to its rapid adaptation in synaptic plasticity, comprising long-term potentiation (LTP) and long-term depression (LTD) [22].

Exocytosis is the process by which eukaryotic cells can release substances into the extracellular space and deliver proteins or lipids to the plasma membrane through the fusion of cargo vesicles. During this process, vesicles (e.g., Golgi-derived vesicles, recycling endosomes, and lysosomes) are transported through microtubules and actin tracks toward specific and polarized areas of the plasma membrane, where vesicles are tethered before fusion to the plasma membrane. Golgi-derived vesicles are docked and tethered by the exocyst complex [23,24,25]. The exocyst is an evolutionarily conserved complex composed of eight subunits (Sec3, Sec5, Sec6, Sec8, Sec10, Sec15, Exo70, and Exo84) [26,27,28] first identified in yeast and then characterized in multicellular organisms. Many cellular functions have been associated with this complex; these functions include cellular growth, division, migration, signaling, cell–cell contact, exo/endocytosis of membranes and proteins, morphogenesis of tissues, and cell invasion of fungi, plant, yeast, and mammal organisms (reviewed in [29]).

Here, we review the exocytic trafficking of glutamate receptors in dendrites with a particular interest in the emerging role of the exocyst complex on constitutive and induced trafficking/tethering of glutamate receptors into the plasma membrane. First, we will briefly summarize basic principles of the secretory pathway in neurons and then particularly the secretory pathway of ionotropic glutamate receptors to establish basic concepts before the discussion of the evidence for the exocyst’s role in the route.

## 2. Membrane Trafficking of Ionotropic Glutamate Receptors: From ER to Plasma Membrane

### 2.1. Membrane Trafficking in Neurons

The endoplasmic reticulum (ER) is the primary organelle where the exocytic trafficking of proteins and lipids begin. In neurons, the ER is spread across the somato-dendritic compartment and axons [21,30]. Interestingly, the ER is also present in dendritic spines and runs in parallel to dendrites as smooth ER (SER) with few ribosomes attached [2,31].

Once proteins are fully folded, both secreted and transmembrane cargoes are trafficked to the ER–Golgi intermediate compartment (ERGIC) and Golgi apparatus for further processing [32,33]. The ERGIC compartment is also present in dendrites showing a tubule-vesicular morphology [34] in contrast to the Golgi apparatus, which is present nearby the soma. Other structures called Golgi outposts are present in some distal dendrites [30] and non-conventional Golgi structures in dendrites, indicating possible satellite structures that sustained the secretory pathway in dendrites and spines [35].

Exocytic trafficking in neurons occurs in two ways: the canonical pathway, where newly synthesized cargoes are delivered to perinuclear Golgi, and a local pathway, where newly synthesized cargoes are translated in dendritic ER membranes and then trafficked to satellite Golgi outposts where they are sorted [21,36]. Even though the Golgi apparatus is not present in all dendritic shafts, the secretory pathway occurs in dendrites in a Golgi-independent secretory pathway. The recycling endosomes have been proposed to participate in the anterograde trafficking of membrane proteins, which has been probed at least for GluA1 subunit AMPARs [37].

Once vesicles are released from Golgi, they are directed to their target membrane using the microtubule cytoskeleton and Rab GTPases [38,39]. Tethering factors mediate the process of docking, and to date, the exocyst complex, as well as other proteins, have been identified in the fusion of secretory vesicles with the plasma membrane [40]. Soluble NSF attachment protein receptor (SNARE) complexes mediate the fusion of vesicles with the target membrane. In the postsynaptic compartment, the SNARE proteins are far less described compared to presynaptic terminals, although in the last ten years, they have been described for glutamatergic synapses. Diverse studies have indicated the presence and function of Syntaxin in dendritic membranes, including Syntaxin-1 (the same SNARE for synaptic vesicle exocytosis), Syntaxin-3, and Syntaxin-4 [41,42]. SNAP proteins have also been described to play a role in exocytic trafficking in neurons, including SNAP-23, 25, 29, and 47 [41,43]. On the other hand, as a v-SNARE, the postsynaptic VAMP2 has been indicated to be part of the AMPAR’s trafficking machinery [44]. Importantly, a few years ago, it was reported that the Syntaxin-4–SNAP2-25–VAMP1 SNARE complex is involved in NMDAR trafficking in basal conditions [45]. Therefore, it is clear that presynaptic and postsynaptic compartments share to some extent the SNARE machinery for exocytic trafficking [41,44,45].

### 2.2. Membrane Trafficking of Ionotropic Glutamate Receptors

As a result of the relevance in physiological and pathological events, the traffic of glutamate receptors at the synaptic place has been intensively studied, but there is much less knowledge about the mechanism underlying the previous steps in the secretory pathway, including sorting. Glutamate receptors are composed of various subunits, such as GluA1-4 for AMPARs and GluN1-3 for NMDARs [46], and their trafficking in and out of synapses is one of the principal mechanisms for rapid changes in the number of functional receptors during synaptic plasticity.

The subunits of AMPA and NMDARs are synthesized in the endoplasmic reticulum (ER) located in the somatic compartment, but GluA1 and GluA2 of AMPARs subunits can be synthesized in dendrites in an activity-dependent manner [47,48,49]. In fact, GluA2 containing AMPARs traffic from dendritic ER to the plasma membrane, where there are pre-existing AMPARs [50]. AMPARs subunits assemble into the ER in dimers firstly, and then in dimers to dimers forming the final tetramer structure of the channel [51], where the N-terminal domain plays a critical role, and the assembly between subunits is not random [52,53]. Similar assembly has been suggested for the NMDARs, where the homodimers GluN1 and GluN2 are required to assure the formation of the tetramer [54,55] and GluN1–GluN2 heterodimers were suggested, too [56].

At the regulatory level, after their synthesis, AMPARs subunits retention at the ER is a common mechanism of quality control, ensuring that unassembled or otherwise defective proteins do not reach the cell surface [57]. NMDARs showed an ER retention motif RXR-type on the C-terminal tail of GluN1. Furthermore, some splice variants lack this retention motif, and their exit from the ER is regulated by PKC phosphorylation [58]. An essential factor in the assembly of AMPARs is the N-glycosylation, where the Asn 63 and 363 are critical for the delivery of the receptor through the secretory pathway [59]. In NMDARs, two sites of glycosylation have been identified in the GluN1 subunit, Asn 203 and 368, for the membrane delivery of NMDARs. Glycosylation sites in the GluN2 subunit do not alter the trafficking of the receptors [60], and three Asn are essential for the delivery of GluN3 subunits in hippocampal neurons [61]. Another mechanism for regulating the surface expression of AMPA and NMDA receptors is the palmitoylation. AMPARs are palmitoylated in cysteine residues at the cytosolic face of Golgi membranes. While the palmitoylation of specific residues occurs at the steady-state and it did not affect the surface expression of the receptor, the palmitoylation of other residues retains the receptors on the Golgi apparatus, and the depalmitoylation release them to the plasma membrane [62]. Two cysteine clusters have been described in the C-terminal tail of GluN2A and GluN2B to be palmitoylated that retained the receptors in internal stores [63].

Protein interactions are also important determinants in glutamate receptor trafficking. One example is the protein LARGE. Mutations in this protein are associated with muscular dystrophy that includes intellectual disability. The LARGE protein is a resident of the Golgi apparatus and retained AMPA receptors in their delivery to the plasma membrane [64]. AMPAR and NMDAR subunits interact with scaffold proteins such as SAP102 through the known PSD95/Dlg1/ZO-1 (PDZ) domain in the ER, and this is required for the subunit to leave the ER [58,65,66,67]. This is consistent with the distribution of SAP102, which, in addition to being present at the synaptic zone similar to PSD-95, is also distributed throughout the cytoplasm in which it may interact with various organelles, including vesicles destined to the postsynaptic membrane [68,69]. The scaffold molecules are not only stabilizing elements but also molecular determinants of receptor exchange and sorting between subcellular compartments. This is supported by findings that some postsynaptic proteins involved in the synaptic clustering of NMDARs and/or AMPARs, such as PSD-95 [70], stargazin [71], SAP97 [69], and GRIP/ABP [72,73], are associated with these receptors as they traffic toward or from the neuronal surface. The transport of AMPA and NMDARs occurs in vesicular organelles, and it depends on the microtubular cytoskeleton that runs along the dendrites [74].

AMPARs require different proteins to be transported through the cytoskeleton. Transmembrane AMPAR regulatory proteins (TARP) is a family of proteins that regulate AMPAR trafficking and gating. Among them, γ-2 (or commonly named stargazing), γ-3, γ-4, and γ-8 are the ones that regulate AMPAR trafficking through direct binding [75]. TARPγ-2 regulates the surface availability of GluA2 in cerebellar cultures [76]. TARPγ-2 also regulates synapse formation and GluA4 dendritic trafficking into synapses in granule cells [71]. Kato and colleagues found that TARPγ-2 enhanced GluA1 and GluA2 cellular trafficking toward the plasma membrane using CHO cells as a model for trafficking assessment [77]. TARPγ-8 loss is related to a dramatic reduction of extrasynaptic AMPAR and a slight decrease in synaptic AMPAR suggesting that indeed, the receptors are trafficked by TARPγ-8 mainly to extrasynaptic sites [78,79]. Furthermore, it seems that there is a possibility that functional redundancy exists within TARPs, since a double knockout of TARPγ-2 and TARPγ-4 completely abolishes AMPAR-related synaptic transmission in cerebellar Golgi cells, while the loss of both proteins individually does not show the same outcome [80]. Perhaps TARPs share their function in AMPAR traffic toward synapse, and this may rely on the cell type that is being studied. In fact, functional compensation has been suggested to occur very commonly between these TARPs due to how closely related they are [81]. For a further review of TARPs on AMPAR trafficking, please see [82].

Cornichon proteins (CNIHs) are also AMPAR auxiliary proteins responsible for its trafficking and gating. CNIH2 overexpression increased AMPAR surface expression in transfected heterologous cells [83] by promoting the export from ER [84]. CNIH2 and 3 bind preferentially to GluA1 subunits; in doing so, they modulate the dendritic transport of GluA1/A2 heteromers to the targeted plasma membrane [85]. Data also show that in CNIH2 and 3 conditional knockout mice, immature AMPARs are retained in the ER, which is consistent with the report of Harmel and colleagues [85]. 

Rab GTPases are essential for glutamate receptor trafficking. Rab39B has been identified to regulate AMPAR exit from the ER and to regulate GluA2 surface availability in non-simulated neurons [86]. Rab11 mediates the translocation of GluA1 into dendritic spines, while Rab8 is responsible for its trafficking within the spine, and both Rab8 and Rab11 are important modulators in LTP-induced trafficking [87,88]. Rab4 play a critical role in spine maintenance during basal conditions, suggesting that constitutive trafficking is a part of Rab4 function [45,88]. Furthermore, Rab5 and Rab10 are part of the endocytosis machinery that direct AMPARs into early endosomes [89], whilst Rab4 and Rab11 redirect AMPAR-containing vesicles toward the plasma membrane during recycling [45,89] (recently reviewed by Hausser [89]). Regarding NMDAR, Rab3 interacting protein isoform 1 (RIM1) has been indicated to localize in dendritic spines where it plays a crucial role in the insertion and recycling alongside Rab11 [90]. This evidence expands the spectrum of functions of RIMs beyond the presynaptic vesicle cycle.

Both receptors have two options of being transported toward the plasma membrane. One is the canonical secretory pathway at the soma of neurons, and the other is by being transported through ER–ERGIC–Golgi structures in dendrites (Figure 2). It has been suggested that NMDARs continue through the secretory pathway, bypassing the somatic Golgi apparatus, being trafficked to dendritic Golgi outposts alongside adaptor proteins SAP97 and CASK [49], illustrating the complexity of the exocytic route for these receptors. In the synapses, glutamatergic receptors could be either directly exocytosed at the synapse or first exocytosed into the extrasynaptic membrane, which is followed by their lateral diffusion at the neuronal surface and trapping at synaptic sites [91].

### 2.3. Membrane Trafficking in Synaptic Plasticity

Changes in synaptic strength are thought to involve the rapid movement of AMPARs into and out of synapses. Several studies have suggested that AMPARs, unlike NMDARs, rapidly and constitutively cycle between intracellular stores and the cell surface [51,92]. This cycling may allow for fast regulated changes in AMPAR numbers in the synaptic zone [51]. Part of the molecular components of these cycling is the VPS35 component of the retromer complex [93] and SorCS1, which is a sorting protein between early and recycling endosomes [94].

Considerable evidence indicates that protein kinases and phosphatases play essential roles in the synaptic plasticity (LTP and LTD; [91]). Some kinases such as the CaMKII and Fyn are thought to mediate directly the signals leading to LTP, whereas others such as protein kinase A (PKA) might play a modulatory role [95]. In the case of LTD or homeostatic plasticity, the output of AMPARs and NMDARs from the synaptic zone needed the activity of phosphatase such as Striatal-enriched protein tyrosine phosphatase (STEP) and Phosphatase and tensin homolog (PTEN), which dephosphorylate specific subunits of these receptors and induce the internalization by a Clathrin-dependent mechanism [96,97].

Surface insertion of the GluA1-subunit of AMPARs is slow in basal conditions and is stimulated by NMDARs activity [98], whereas GluA2-subunit exocytosis is constitutively rapid. During the insertion of AMPARs in LTP, the vesicles from the recycling endosome fuse with the membrane with the help of synaptotagmins 1 and 7, which are vesicle proteins responsive to calcium, as occurred in the presynaptic compartment with neurotransmitter vesicle exocytosis [99]. In this regard, AMPAR is exocytosed using the SNARE complex formed by Syntaxin-3, VAMP2, and SNAP47 [100]. However, Syntaxin-4 has also been suggested to regulate AMPAR trafficking during LTP [42,101].

It is important to consider that the synaptic activity and the changes in the levels of AMPAR regulated by their recycling follow different signaling pathways. It has been demonstrated that in non-stimulated synapses, the majority of AMPAR are recycled via the Arf6 protein and in stimulated synapses via the transferrin receptor. In this context, TC10 participates in the recycling of AMPAR together with Arf6, particularly in those stimulated or strengthened synapses [102]. Through interaction with the GluA2 subunit, the GRIP1 scaffold protein regulates the presence of GluA2 at synapses, which is an activity-dependent recycling pathway [103]. Upon internalization, AMPARs are sorted within early endosomes either to a specialized recycling endosome compartment for reinsertion to the plasma membrane or late endosomes and lysosomes for degradation [104]. This recycling process has recently been described to be dependent on the endosomal protein GRASP1 [105] and synaptotagmin 3 [106].

The synthesis and exit of the NMDAR subunit from the ER can be altered by the neuronal activity [36]. The consequences of this activity-dependent switch are important because the neuronal activity can enhance the exit of newly synthesized NMDARs from the ER, thereby increasing the secretory trafficking and surface delivery of NMDARs or favoring the retention in the ER [36]. The constitutive internalization of NMDARs is much lower than that of AMPARs. Thus, NMDARs are considered to be relatively stable once delivered to the neuronal surface. However, early in development, the NMDARs constitutive internalization velocity is similar to the AMPARs, but this constitutive endocytosis of NMDARs is lost in mature neurons [107]. NMDARs can be removed from the synaptic zone either by endocytosis directly or by lateral movement within the plasma membrane away from the synapse, which is followed by endocytosis through a clathrin-dependent mechanism. In addition, clathrin-coated pits can internalize both AMPA and NMDARs at extrasynaptic sites not associated directly with synapses [108].

## 3. The Exocyst Complex

The exocyst, as expected for a protein complex, presents many direct interactions between the subunits in yeast and mammals. Therefore, these interactions could participate in the activity of the complex. As a result of this, interactome studies became necessary for every organism mentioned. Positive strong interactions were found between Sec15 and Sec10, Sec8 and Sec10, Sec5 and Sec6, Sec6 and Exo70, Sec3 and Sec6, and Sec3 and Sec8. In addition to these strong interactions, numerous weak interactions were also detected, including Sec5 and Sec15, Sec6 and Sec5, Sec8 and Exo84, Exo70 and Sec5, Exo70 and Sec8, among others [109]. The existence of these weaker interactions brings up the possibility that the stability of the intact complex may be achieved through a series of interactions that are not easily detected by pair-wise interaction analysis. Studies done until now have revealed similar interactions between subunits in both yeast and mammalian exocyst [109,110,111,112,113,114], but its functions may vary. Interactions of the exocyst with its regulators have been shown to differentiate cell types, including Sec3-Rho1p, which is not observed in mammal and TC10-Exo70, which is specially related to polarization in neuronal cell types including PC12 (pheochromocytoma 12) and hippocampal neurons. Additionally, it has been proposed that the exocyst complex uses the same basic proteins to produce its functions, but the regulators acting toward the exocyst are different in the signaling context [109,110,115].

Several reports have intended to elucidate the morphology of the complex by different approaches [116,117,118,119]. The structure of the exocyst complex has been solved by cryogenic electron microscopy in its ensembled state in yeast. Modeling of this cryo-EM data showed that the exocyst consists of long curved rods (exocyst subunits) that pack against each other to form a structure measuring 320 Å long and 130 Å wide [119]. This rod-like structure of the exocyst subunits grants the ability to interact closely with each other through helix bundles in an antiparallel fashion [120]. However, the best proposal so far is developed by Picco and colleagues [118]; they used a model which serves to understand the localization and function of the exocyst in vesicle tethering to the plasma membrane also in yeast; however, no study has reported the morphology of the complex in animals. In addition, it is important to mention that the exocyst exists as two subcomplexes in many organisms, including yeast [110], plants [121], and mammals [111,114]. These subcomplexes comprise four subunits each; subcomplex 1 (SC1) comprises Sec3, Sec5, Sec6, and Sec8, and subcomplex 2 (SC2) comprises Sec10, Sec15, Exo70, and Exo84. The interaction between SC1 and SC2 triggers the activity of the complex in exocytic processes [110,122]. The stability of the complex depends on the presence of all subunits. The silencing of Sec8 results in disassembly of the SC1 but not the SC2. Conversely, the silencing of Sec10 disassembles SC2 but not SC1 [114]. Both of these silencings evoke a reduction in complex and subcomplexes functionality by reducing its proximity to the plasma membrane. The inhibition of endogenous Exo84 (SC2) reduced the interaction between Sec10 (SC2) and Sec6 (SC1), resembling complex partial dissociation [123]. Copping Sec3 in mitochondria also affects the exocyst’s assembly in vesicle tethering at the plasma membrane [124]. Finally, most of the exocyst subunits are essential for maintaining the octameric complex; a loss of Sec5, Sec6, Sec8, Sec10, Exo70, and Exo84 resulted in the splitting of the octameric complex into SC1 and SC2 due to the increased instability of both subcomplexes [110,125].

Using all the data available, we present a cartoon model of the exocyst comprehending the interactions between subunits and possible localization in vesicle fusion sites where vesicles are tethered by the exocyst complex (Figure 3). In this model, Sec3 and Exo70 target the exocyst to the plasma membrane, as expected [126]. The opposite positioning of Sec10 and Sec15 is observed compared to Sec3 and Exo70; this positioning is essential to the contact of the exocyst with the vesicles where Sec15 interacts with several Rabs [127] to be its effector protein [128]. Particularly, Sec15 is a known interactor of Rab11 to deliver recycled vesicles to the plasma membrane [127,128,129]. Although not all possible interactions of Sec15–Sec10 with Rabs have been tested, Rabs small GTPases present interesting participation in exocyst-mediated exocytosis, being a bridge between the exocyst complex and vesicles to be delivered to the plasma membrane.

The dynamics of the exocyst complex during exocytosis is underway to understand how the complex works to deliver vesicles to the plasma membrane. Early research on this topic proved that the majority of the complex (except Sec3 and Exo70) ride the vesicles along actin toward the exocytosis sites in yeast [126]. In addition, in yeast, Donovan and Bretscher showed that vesicles arrive at the plasma membrane 17 s before membrane fusion with the exocyst and that Sec3, Sec5, and Sec15 leave the vesicles at the same time that fusion occurs [130]. In mammals, the dynamics seem to be slightly different. Sec8 arrives at the membrane with the vesicle approximately 12 s before membrane fusion, and it remains docked 2 s after fusion in HeLa cells [131]. Recently, Ahmed and colleagues have shed light on the dynamics of the exocyst in a subunit-related manner. They confirmed that the exocyst complex arrives at the plasma membrane with vesicles 10–15 s before fusion. Sec5 and Sec8 arrive together and Exo70 arrives at 77 milliseconds after Sec5 [114]. In addition, Exo70 leaves 1.3 s after fusion, while Sec3 leaves at 0.4 s before fusion, indicating the separate roles of these exocyst subunits in membrane targeting and fusion [114]. Future work remains to be done in order to elucidate more precisely the dynamics on different cell types, including polarized and non-polarized cells.

The exocyst, as expected for a complex involved in polarized exocytosis, is localized to specific regions of the plasma membrane. In multicystic dysplastic kidney (MCDK) epithelial and pancreatic acinar cells, Sec6 and Sec8 subunits were found to have both perinuclear and plasma membrane enrichment [132,133]. Precisely, membrane-localized exocyst staining was found in tight junctions [133], where it delivers basolateral membrane and proteins through interacting with the Par3/6 complex [122]. Sec10 also present a similar enrichment [134]. When cell–cell contacts were disrupted in MDCK cells, plasma membrane-localized Sec8 redistributed into the cytoplasm [135], suggesting that the membrane localization of Sec8 is dependent on cell polarity. Sec6 and Sec8 subunits colocalized with cargo vesicles, and the inhibition of exocytosis of these vesicles disrupted the recruitment of the exocyst to the plasma membrane, suggesting that the exocyst is present between the trans-Golgi network and plasma membrane, and its regulation controls where the pool of complex are localized [133,135]. The inhibition of Sec8 by antibodies has been found to disrupt plasma membrane protein transport and membrane addition in MCDK cells [132,135,136]. Consistently, the inhibition of Sec6, Sec8, and Exo70 subunits by RNA interference (RNAi) exhibited a perturbation in the exocytosis of GLUT4 (Glucose transporter type 4) to the plasma membrane in adipocytes [137]. Additionally, the overexpression of Sec6, Sec8, and Sec10 mutants resulted in diminished insulin secretion in pancreatic cells and disrupted neurite outgrowth in nerve growth factor (NGF)-differentiated PC12 cells because the mutants work as dominant-negative interfering with the normal function of the complex [138,139]. In undifferentiated PC12 cells, Exo70 was found in perinuclear localization, and upon differentiation with NGF to promote neurite formation, Exo70 was found distributed between perinuclear and growth cone region [139]. Immunostaining of Sec6, Sec8, Sec15, and Exo84 showed similar localization [140]. In mammals, the maintenance of cell polarity is achieved by basolateral membrane addition delivered by the exocyst complex [115,141,142]. Interestingly, all exocyst subunits exhibit a microtubule-like localization [143]. In NRK cells, Exo70 by itself can disrupt microtubule formation in vitro, and consistently, its overexpression destabilized the microtubule network near the plasma membrane [143]. Moreover, a localized destabilization of microtubules promotes plasma membrane addition at these sites in hippocampal neurons [144]. An interesting function in autophagy has been related to the exocyst as well. Sec5 controls the autophagy influx of EGFR on starvation conditions [145]. Exo70 mutants also disrupt membrane trafficking in autophagy-related vesicle transport [121], which has been related to the ancestral association between the exocyst and autophagy-related Atg8 proteins [146]. Indeed, the exocyst role in autophagosome biogenesis through Atg9 trafficking regulation has been reported [147]. Finally, it has been proposed that the exocyst acts as a tethering complex for the autophagic complex on autophagosome formation [148]. Thus, the exocyst localizes at specific sites and has relevant functions in exocytosis, membrane expansion, and autophagy in different cell types, providing the molecular machinery to maintain these processes optimally.

All the topics discussed above, in non-neuronal and neuronal cells, denote an increasing interest in the function of neuronal exocyst complex that may present possible roles in synapse formation and maintenance due to the addition of membrane and insertion/recycling of glutamate receptors.

### Exocyst in Neurons

In neurons, the exocyst has been identified in neurite growth cones [139] and multiple domains throughout the axon in equidistant patches [149]. Remarkably, these patches observed in the axon appear before synaptogenesis, suggesting that the exocyst participates in synapse formation. Gaining knowledge on this topic has been difficult, since knockout animals for Sec5, Sec6, and Sec8 are lethal. Sec5 and Sec6 knockout in Drosophila is deadly in the larvae state due to membrane and protein addition impairment in neuromuscular junctions [150,151]. Although Sec5 knockout is deadly in the larvae growth period, the null mutant does not alter synaptic transmission before, suggesting that Sec5 is necessary for exocytosis of Golgi-derived vesicles and is not required for exocytosis of the synaptic vesicle [150]. The only murine knockout is Sec8; this model presents a lethality at embryonary state E7.5 before the neuronal differentiation [152]. Moreover, a mutation in Sec15 is not lethal, and neurons extend neurites normally but fail to select their synaptic partner at excitatory synapses [153]. An impairment of neuronal cell polarity and neurite outgrowth is achieved in the absence of native exocyst subunits in several cell types, including primary neurons, PC12 cells, or multicellular model organisms [139,150,154,155]. Drosophila primary neurons lacking Sec5 and Sec6 exocyst subunits show an impairment in dendrite growth and similarly occur in *C. elegans*, where the deletion of Sec6 and Sec8 diminish the arborization of neurons. In rat hippocampal neuron cultures, knocking down Exo70 also has an impairment in dendrite growth, and its overexpression produces the opposite effect, increasing the development in dendritic processes [156]. In developing neurons, the exocyst is highly enriched in axon growth cones and filopodia, where it participates in membrane extension at the most distal end of the axon [149,157]. Interestingly, Exo70 has been related to filopodia formation in several cell types, including neurons [156,158,159,160,161]. Unlike axons, dendrite growth is not restricted to direct tip extension; instead, it may relay in distal exocytosis mediated by exocyst subunits, suggesting that membrane addition in neurons might be more complicated than in other cells. In fact, exocytosis sites might be located in branch points where the exocyst is localized [154,162,163]. This no-tip exocytosis model was seen in Drosophila as well, where it has been shown that the exocytosis in branch or distal points of the dendrite may not be neuron-specific [129].

It has been suggested that the exocyst is involved in receptors/transporters trafficking and exocytosis. For example, Exo70 is required to insert the insulin-like growth factor 1 (IGF1) receptor into the axonal growth cone membrane by promoting membrane expansion through the Rho GTPase TC10 [157], and GTP-TC10 associated with Exo70 in the vicinity of the plasma membrane promotes the release of Exo70, allowing vesicular fusion [164]. Additionally, Exo70 participates in membrane expansion and the growth of axons in developing neurons [157], suggesting that it may provide the sites where membrane addition is necessary. Sec8 also interacts with PSD-95 to be delivered to the dendritic spine. This interaction is regulated by cypin, suggesting a possible regulator to deliver PSD-95 to the synapse [165]. Sec3 modulates the availability of Glycine transporter (GLYT1) in the membrane and probably modulates NMDAR activity due to the Sec3-dependent delivery of GLYT1 to synaptic sites, which maintain the optimal working concentration of glycine near NMDAR localization [166]. Sec5 is responsible for postsynaptic membrane growth in the Drosophila neuromuscular junction; and in mice, it is accumulated in a postsynaptic compartment in an RalA/activity-dependent manner, suggesting that the regulation of a neuronal exocyst complex might be due to neuronal activity in cultured hippocampal neurons [167]. From a synaptic transmission point of view, it has been demonstrated that the exocyst is not involved in neurotransmitter release [150]. Drosophila null mutant of Sec5 presents an impaired transport of integral membrane protein to the cell surface, and concomitantly, neurite outgrowth is reduced, which is probably due to impaired membrane addition. Nevertheless, synaptic transmission persists [150]. Interestingly, the knock-down of Sec10 in pre and postsynaptic compartments also presents a similar outcome in synaptic transmission, and when synaptogenesis was analyzed, the results showed that Sec10 also does not participate in this process [168]. In rats, Exo70 apparently is not involved in synaptogenesis as well, although it was evaluated in a different type of synapse in the calyx of Held in the auditory central nervous system. In this giant synapse, Exo70 dominant-negative produced structural changes in the calyx, while the synaptic transmission was not impaired [169]. This is intriguingly surprising, because we have related Exo70 to synaptogenesis and synapse stabilization in rat hippocampal neuron cultures [156]. Exo70 overexpression increases excitatory synapse formation in developing neurons, and knocking down Exo70 expression produces the opposite effect, reducing synapse formation. Synaptic puncta changes also correlate with the mature-induced state of dendritic spines due to Exo70 overexpression [156]. It seems that the molecular machinery present in the calyx of Held and hippocampal synapse are very similar, containing Piccolo [170], RIM1 and 2 [171], synapsin [172], and AMPA receptors, among others [173]. Structurally, presynaptic active zones and postsynaptic densities are very similar between both synapses; the one major difference is that calyx of Held harbor several hundreds of small active zones within one giant terminal [173]. Thus, synaptogenesis might have different cellular cues to be carried out properly between cell types. This might be the reason why Exo70 behaves in a cellular-dependent manner. Finally, in mice hippocampal slices transfected with Exo70 or its dominant-negative (DN), AMPAR-evoked postsynaptic currents (EPSC) were diminished only when Exo70 DN was expressed, leaving NMDAR EPSC unchanged. On the other hand, the transfection of Sec8 DN impairs both AMPAR and NMDAR EPSC [174].

## 4. Exocyst in Ionotropic Glutamatergic Receptor Trafficking and Delivery

When it comes to the exocyst—a complex related to traffic, recycling, and exocytosis—several reports have intended to elucidate how it may function into the delivering and recycling of AMPA and NMDA receptors. First, it is important to note that interactions between the exocyst and AMPA/NMDA receptors have been detected both in vivo and in a heterologous cell system. Positive interactions in brain and synaptosome preparation were detected between Sec8, Sec6, and GluN2B [68]. In addition, Sec8 and Exo70 interact with GluA1-3 and GluN1 [174]. This interaction occurs early in the ER, where the exocyst recruits newly synthesized receptors to be ridden through actin cables [68], and this interaction is essential to the delivery of the receptors to the plasma membrane. Blocking this interaction through the Sec8-PDZ domain deletion resulted in cell surface delivery impairment. GluN1 and GluN2B trafficking toward the cell surface are carried out in a trimeric complex, which includes SAP102, Sec8, and the receptors in heterologous cells and neurons [68,175]. Reaching the plasma membrane, Sec8 controls the directional movement of AMPA receptors to the synapse through PDZ domains, and Exo70 mediates the insertion of the receptor to the plasma membrane. An interference of Exo70 function with Exo70 DN lacking a C terminal domain results in the accumulation of AMPA receptors within the spine, forming an associated complex that has not been fused to the plasma membrane [174]. This DN impairs the delivery of GluA1 to the dendritic spines surface in cultured hippocampal neurons, suggesting a synaptic transmission malfunction. Indeed, Exo70 DN reduced the AMPAR excitatory postsynaptic currents, while Sec8 DN affected both AMPAR and NMDAR currents [174]. In Gerges and colleague’s work, they conclude that the exocyst complex—through Exo70—mediates AMPARs insertion directly in the synaptic zone rather than at the extrasynaptic zone. This event is at least present in basal transmission, since non-stimuli were delivered.

## 5. Glutamatergic Receptor Trafficking: When and Where to Choose the Exocyst

To date, it is not totally understood in which neuronal context the exocyst participates in glutamatergic receptor trafficking and exocytosis. In this section, we will discuss the constitutive and stimulated trafficking of AMPA and NMDA receptors through the exocyst complex.

As mentioned above, AMPARs and NMDARs ride along microtubules in tubulovesicular structures in dendrites [68,176] using kinesins as motor proteins [177]. AMPA predominantly prefers to be transported through dendrites by GRIP1 [178,179], and interfering with its expression or interactions severely diminishes AMPAR trafficking [82,178,180] (for a further review, see [82]). Although GRIP1 is considered to be the first option for AMPAR trafficking, Sec8 also has been related to AMPAR trafficking. By co-expressing GluA2 and Exo70 or Sec8 DN in neuronal cultures, Gerges and colleagues showed that GluA2 trafficking into distal dendrites was normal when Exo70 DN was expressed, indicating that AMPARs transport along dendrites does not require a fully functional Exo70 subunit. On the other hand, the dendritic transport of GluA2 is severely impaired when Sec8 DN was expressed [174]. Sec8 DN lacks its C-terminal domain, which contains the PDZ domain, allowing its interaction with AMPAR, and thus, it is believed that GluA2 is trafficked through the Sec8 PDZ domain. In addition, Gerges and colleagues found that expressing Sec8 DN caused a reduction in AMPAR EPSC, supporting its involment in trafficking [174]. Interfering with GRIP1 and Sec8 individually is sufficient to affect AMPAR trafficking similarly; therefore, it is plausible to hypothesize that they might be in the same transport complex. Of note, GRIP1/Sec8 interaction has been observed in GluA2 recycling after stimulation with NMDA [103]. This hypothesis gives the exocyst a more relevant role in AMPAR trafficking alongside GRIP1 through dendrites. Future experiments should address this hypothesis.

NMDAR is transported in compartments containing EEA1 [176]. Its trafficking to the cell surface employs a large protein complex containing GluN1, GluN2B, SAP102, mPins, and Sec8 [68,181,182]. As discussed above, NMDAR travels through dendrites bound to SAP102, which in turn binds to Sec8. Both NMDAR and Sec8 bind SAP102 in the same region containing PDZ domains [68]. Using an Sec8 DN that lacks the PDZ-binding domain, they showed that Sec8–SAP102 binding was blocked, thus preventing the delivery of NMDARs to the cell surface. This provides evidence that NMDAR surface delivery needs both Sec8 and SAP102. Then, they deleted the PDZ-binding domain of GluN2B and found that this subunit can be delivered to the cell surface by a Sec8/SAP102-independent mechanism [68,183]. The above complex also binds Sec6 and Exo70, suggesting that some or the entirety of the complex travel with the ensembled NMDAR toward the cell surface and synapse [68]. Therefore, up until now, we know that NMDAR trafficking is carried out by the exocyst and that there must be another mechanism of transport that has not been discovered yet.

AMPAR and NMDAR have been shown to participate in LTP-related processes (for a further review, see [184,185,186]). Studies of AMPAR trafficking in synaptic plasticity have focused on the idea that LTP expression depends on alterations in AMPAR number at synapses. This conception lays out evidence suggesting that (1) AMPARs are inserted in the extrasynaptic membrane without stimulus and after LTP induction, the receptors rapidly diffuse laterally into synapses [187,188], and (2) AMPARs are rapidly inserted into the synapse upon LTP induction [189,190]. The first involves constitutive exocytosis by which AMPARs are inserted into the membrane, and the second requires a pool of AMPAR-containing vesicles ready to be inserted into the plasma membrane upon stimulation. We know that constitutive trafficking toward the plasma membrane and delivery of AMPAR is exocyst-dependent [174]; in this context, the exocyst provides the machinery for the vesicular delivery of AMPAR without stimuli. On the other hand, AMPAR-containing vesicles must be delivered to the plasma membrane before fusion in their exocytosis both in a basal and LTP context. The exocyst likely provides the molecular machinery to deliver the vesicles in an LTP context. This is supported by the fact that the exocyst is considered as the only complex acting in vesicular tethering to the plasma membrane in many cell types [24,191,192]. It is important to note that the exocyst is finely regulated by phosphorylation in its trafficking and exocytosis function [193,194,195,196,197], and LTP induces several phosphorylation processes that maintain potentiation [185]; in this scenery, crosstalk between those events might exist under physiological conditions. Therefore, it is imperative to investigate whether LTP regulates exocyst phosphorylation and activity to induce rapid AMPAR insertion into the synapse.

To the best of our knowledge, there is no report regarding alteration in NMDAR dendritic trafficking upon LTP induction. However, it is believed that LTP only increases local synaptic trafficking [185], which requires an intracellular NMDAR pool to be delivered into non-synaptic sites. Indeed, LTP induction leads to NMDAR rapid surface expression mediated by PKC and Src family kinases [198,199]. Furthermore, PKC activation induces the rapid lateral mobility of NMDAR into synapses [200], suggesting that neuronal activity regulates NMDAR rapid insertion into synapses. Interestingly, Sec5 synaptic localization has been shown to be regulated by neuronal activity [167] by increasing its positioning on dendritic spines in an RalA-dependent manner. This implies that LTP induction could regulate the exocyst’s synaptic localization and therefore increase NMDAR local trafficking and tethering to the plasma membrane. As discussed before, the exocyst complex is the only complex to tether membranes in exocytosis processes [24,191,192]. Therefore, it might be related to glutamate receptors insertion into the plasma membrane in an LTP context.

## 6. Conclusion and Projections

Membrane trafficking is essential for cellular functioning. One of its processes is the trafficking, tethering, and exocytosis of cargo vesicles between Golgi and the plasma membrane. The exocyst complex is responsible for trafficking and, in particular, the tethering of post-Golgi vesicles in all eukaryotic cells. In neurons, it has been related to neuronal development and synaptogenesis, as well as the availability of receptors within the synapse. Most of the studies have focused on constitutive trafficking and the availability of the receptor at synapses. Giving the fact that LTP needs augmented trafficking and the exocytosis of receptors to develop and maintain potentiation for several hours, it is intriguing that no report has ever investigated these processes regarding the exocyst in an LTP context. The exocyst is finely regulated by phosphorylation and GTPases; thus, regulation by LTP-involved kinases could be a mechanism that neurons use to modulate the trafficking and tethering of receptor-containing vesicles within the synapse (Figure 4).

As discussed in the previous sections, exocyst subunits have individual functions in the trafficking and delivery of cargo vesicles to the plasma membrane. For example, Sec8 controls AMPAR dendritic vesicle trafficking, and Exo70 is involved in tethering those vesicles to the synaptic membrane; therefore, Exo70 regulates receptor availability on the membrane at synapses in final exocytosis processes. Thus, the exocyst’s function is more complicated than previously thought, because the complex does not function in a united manner doing a task; instead, some components have specialized functions that are separate from the rest of the complex. This presents an open field of study on the exocyst’s role in membrane and receptor trafficking to synaptic sites and recycling of the synaptic components.

In addition to the LTP-regulated localization and function of the exocyst complex, there is a lot of attention of our group on neurological disorders and the possibility that the exocyst is involved in them. For example, there is a consensus that the exocyst is involved in developmental disorders, which are strictly related to alteration in trafficking and exocytosis of the membrane and cargos (please see [29] and the exocyst in the neuron section of this review). Furthermore, neurodevelopmental and neurodegenerative disorders comprise trafficking impairments on their own [201,202,203], and thus, it is interesting to study the possibility that the exocyst complex is involved in those diseases. In addition, this could provide newly discovered targets to develop pharmacological treatments.

## Figures and Tables

**Figure 1 cells-09-02402-f001:**
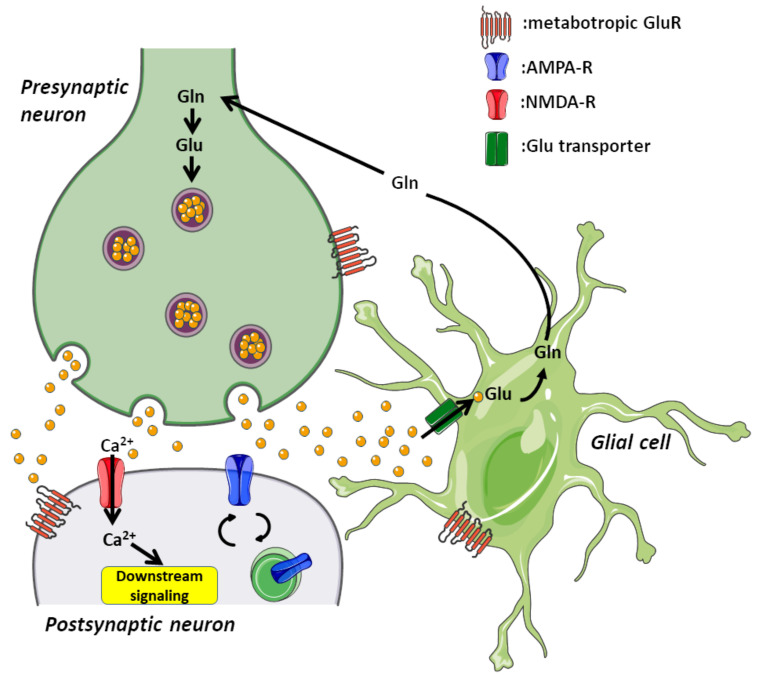
Excitatory synapse structure. Excitatory synapses are the most common in the central nervous system. It comprises a presynaptic neuron, a postsynaptic neuron, and a glial cell. The presynaptic compartment contains the cellular machinery for tethering, exocytosis, and endocytosis of glutamate-containing vesicles; refilling these vesicles is carried out in this compartment as well. In the postsynaptic compartment, we find glutamate receptors and their associated signaling coupled to exocytosis and recycling machinery. Additionally, there are supporting glial cells that maintain this basic synaptic structure. The correct functioning of synapses depends on all three cellular components in order to maintain physiological activity.

**Figure 2 cells-09-02402-f002:**
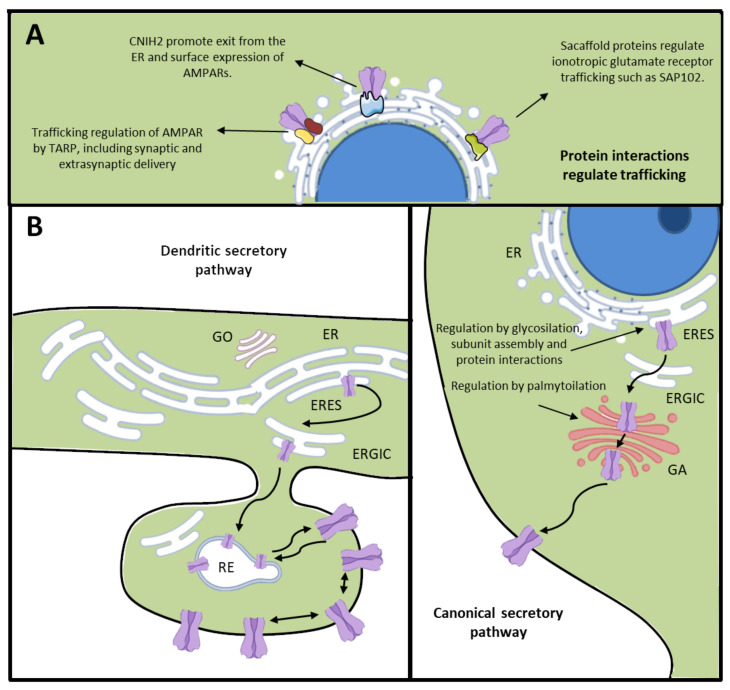
Ionotropic glutamate receptor trafficking. (**A**) Protein interactions regulate exocytic trafficking of ionotropic glutamate receptors, such as TARPs, Cornichon proteins, and scaffolding proteins, among others. (**B**) (Left) Dendritic secretory pathway, indicating the presence of ER, ERES, ERGIC, and GO in the dendrites. Moreover, it shows the action of recycling endosomes in the anterograde secretory pathway for glutamate receptors to the plasma membrane. (Right) Canonical secretory pathway for glutamatergic receptors in the somatic ER, and trafficking through ERGIC and GA. Neurons use both pathways in neuronal development and plasticity. ER: endoplasmic reticulum, ERES: endoplasmic reticulum exits sites, ERGIC: ER–Golgi intermediate compartment, GO: Golgi outpost, GA: Golgi apparatus, TARP: transmembrane α-amino-3-hydroxy-5-methyl-4-isoxazole propionic acid receptor regulatory proteins.

**Figure 3 cells-09-02402-f003:**
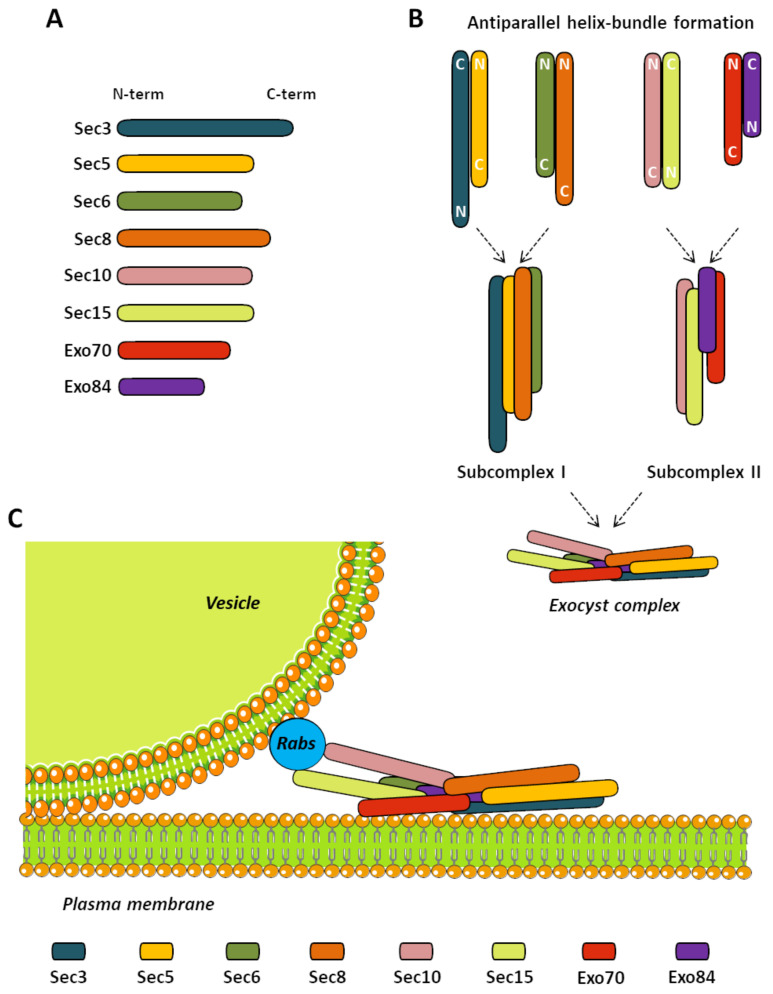
The exocyst complex. (**A**) Bars are examples of the scaled size corresponding to each subunit. (**B**) The scheme depicts antiparallel helix-bundle formation in the assembly of the exocyst complex. Subunit positioning corresponds to an approximation of the interaction experiments discussed in this review. Primary pairs are formed by Sec3–Sec5, Sec6–Sec8, Sec10–Sec15, and Exo70–Exo84. These pairs interact closely to induce the formation of the subcomplexes 1 and 2. Then, both subcomplexes bind together to finally stabilize the functional exocyst complex. (**C**) A cartoon of the exocyst complex is shown. The exocyst acts by tethering secretory vesicles to the plasma membrane in specific sites where the cargo is needed. The model shows a rod-like molecular structure of the subunits. In this model, Sec3 and Exo70 target the exocyst to the plasma membrane. An opposite positioning of Sec10 and Sec15 is observed compared to Sec3 and Exo70; this positioning is vital to the contact of the exocyst with the vesicles where Sec15 interacts with several Rabs to be its effector protein.

**Figure 4 cells-09-02402-f004:**
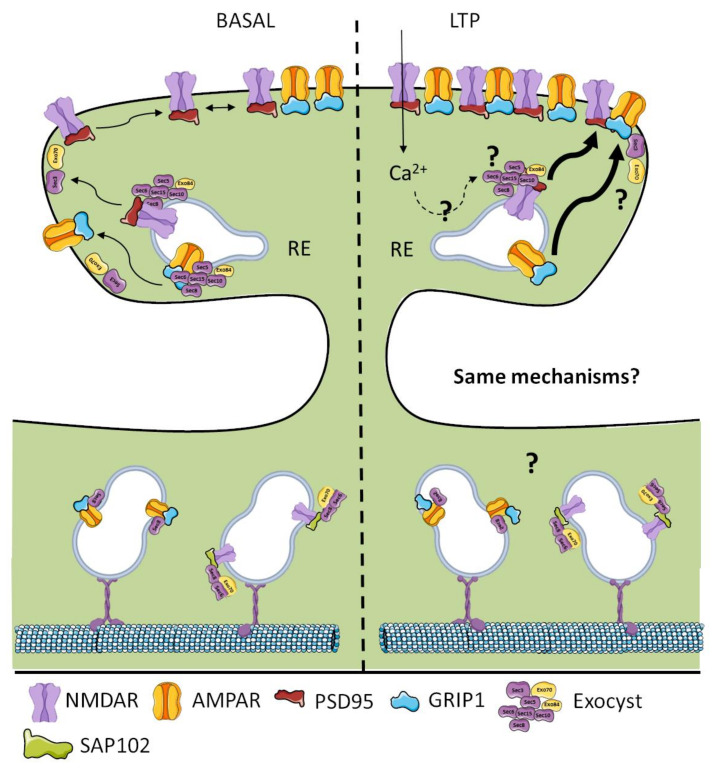
Exocyst in glutamatergic receptors trafficking. (Left) In basal conditions, AMPARs are trafficked along with the GRIP1 scaffold protein and the Sec8 subunit of the exocyst complex. NMDARs, on the other hand, are trafficked in vesicles along with SAP102 scaffold protein and subunits of the exocyst complex such as Sec8, Sec6, and Exo70. (Right) It is believed that the LTP-driven trafficking of glutamate receptors is augmented. In this context, the role of the exocyst complex is unknown. Nevertheless, the exocyst has been suggested as the only tethering complex at the plasma membrane in several cell types. Through its intrinsic exocytic faculties, it would participate in LTP-induced trafficking and the exocytosis of glutamate receptors. LTP: long-term potentiation.

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
