# Peer review of "Glutamatergic Receptor Trafficking and Delivery: Role of the Exocyst Complex"

_cells, 2020, doi:10.3390/cells9112402_

Round 1
Reviewer 1 Report
Review of Cells 930813 “Glutamatergic Receptor Trafficking…” by Lira et al.
Overall, this review is well-written and focuses on an important topic. Some of the background information is treated in a very cursory manner, but overall the review appears to be acceptable.
Moderate grammatical errors and non-standard English usage throughout. We recommend careful editing.
Line 59: trafficking and secretory mechanisms of neurotransmitter vesicles is well studied, but the constitutive and homeostatic trafficking and secretory mechanisms at or near the pre-synapse is much less understood.
Line 70: not all exocytosis is by Golgi-derived vesicles. e.g. exocytosis of recycling endosomes, which is discussed later in the review. But here the authors talk about exocyst only in the context of post-Golgi.
Line 140: palmitoylation is cytosolic, so it does not occur “within” the Golgi, but “at the cytosolic face of Golgi membranes”
Line 160-161: there is a body of literature looking at the roles of Rab8 (together with Rab6) in the secretory pathway (see e.g. Hoogenraad & Akhmanova). I think more should be said about the general functions of Rab8 here; otherwise, you are just listing things and not drawing more meaningful connections within the literature.
Reviews of Rabs cited from 2013-2014 are pretty out of date, especially as it relates to neuro. I suggest citing more recent reviews and recent primary research (e.g. Hoogenraad, Akhmanova) looking at Rab GTPases in secretory trafficking in neurons.
Author Response
- Comment: Review of Cells 930813 “Glutamatergic Receptor Trafficking…” by Lira et al.
Overall, this review is well-written and focuses on an important topic. Some of the background information is treated in a very cursory manner, but overall the review appears to be acceptable.
Answer: We appreciate the reviewer's comment, for us, this is a very important issue, and we consider that with the corrections made to the manuscript, it improved significantly.
- Comment: Moderate grammatical errors and non-standard English usage throughout. We recommend careful editing.
Answer: We carried out a better and careful revision of English, trying to comply with what is indicated in the comment.
- Comment: Line 59: trafficking and secretory mechanisms of neurotransmitter vesicles is well studied, but the constitutive and homeostatic trafficking and secretory mechanisms at or near the pre-synapse is much less understood.
Answer: corrected
- Comment: Line 70: not all exocytosis is by Golgi-derived vesicles. e.g. exocytosis of recycling endosomes, which is discussed later in the review. But here the authors talk about exocyst only in the context of post-Golgi.
Answer: Corrected
- Comment: Line 140: palmitoylation is cytosolic, so it does not occur “within” the Golgi, but “at the cytosolic face of Golgi membranes”
Answer: Corrected
- Comment: Line 160-161: there is a body of literature looking at the roles of Rab8 (together with Rab6) in the secretory pathway (see e.g. Hoogenraad & Akhmanova). I think more should be said about the general functions of Rab8 here; otherwise, you are just listing things and not drawing more meaningful connections within the literature.
Answer: We appreciate the reviewer's comment. At the end of section 2.2. We incorporate more antecedents related to the role of rab8, but also of other Rabs, with the idea of giving a more comprehensive vision of the process described. Line 209-221 in the new version of the manuscript englobes this topic with references 85-89.
- Comment: Reviews of Rabs cited from 2013-2014 are pretty out of date, especially as it relates to neuro. I suggest citing more recent reviews and recent primary research (e.g. Hoogenraad, Akhmanova) looking at Rab GTPases in secretory trafficking in neurons.
Answer: We appreciate the reviewer's comment. We incorporated the following references with the idea of updating the references that we previously selected.
Ref. 44. Gu et al., 2016. DOI: 10.1073/pnas.1525726113. Ref. 88. Hausser et al., 2019. DOI: 10.1080/21541248.2017.1337546
Reviewer 2 Report
In this manuscript entitled « glutamatergic receptor trafficking and delivery: role of the exocyst complex », Lira et al. review a large body of literature on several subjects dealing with neuronal development (neurite outgrowth and synaptogenesis), synaptic transmission and post-synaptic receptor trafficking in neurons. In addition, the authors review the structure of the exocyst, a complex of 8 proteins conserved from yeast to mammals, and its role in various cellular contexts, mainly in the exocytosis of post-Golgi transport vesicles. In the last part, the authors review the involvement of the exocyst complex in post-synaptic glutamate receptor trafficking. Deciphering the role of the exocyst complex, a central player in intracellular trafficking and exocytosis, in the membrane trafficking of post-synaptic receptors, is an important question in current cellular neuroscience. Therefore, the topic of this review should be of great interest to the readership of Cells. However, I find many parts of the review not clear, some information is repeated while other important data is missing. In conclusion, this review needs to undergo major changes before it is suitable for publication.
Below are my specific remarks
- Abstract: only post-Golgi trafficking is evoked but for AMPAR trafficking this pathway is not central. Recycling endosomes (REs) are important for trafficking from the endoplasmic reticulum (ER) and for recycling. This should be mentioned in the Abstract.
Line 23: please replace “superior eukaryotes” by multicellular organisms, or vertebrate/mammalian cells (also line 74)
Line 26: not “vesicle-containing receptors” but the reverse
- Introduction: it gives a succinct overview of the architecture of synapses and its function. In mentioning the diversity of AMPARs (line 55), the kinetics of channel opening, closing and desensitization should be evoked as well. The connection with the exocyst complex with membrane trafficking in neurons should be presented already in this paragraph, perhaps by evoking the unique challenges a neuron has to face in terms of receptor trafficking.
- Part 2. Line 106-108. It is not true that the exocyst is the only tethering factor identified for exocytosis. For synaptic vesicles, Munc18 and 13 play an essential role (see for example review by Brunger et al. 2019). The role of Rab GTPases could be expanded. The part on SNAREs needs to be rewritten. In particular, Jurado et al. 2013 have documented the role of syntaxin3, SNAP47 and VAMP2 in synaptic plasticity. Also, SNAP23 and 25 are not v SNAREs.
- Part 2.2. This part is very difficult to read. An expanded scheme from figure 2 with the proteins at play would help. There is no mention of proteins associated with GluA1-4 (TARPs, CNIH, FRRS1L) which are important for AMPAR trafficking.
Line 159. In ref 66 the colocalisation with EEA1 is only significant in very immature neurons, before synapse formation. It should not be emphasized here, or in Figure 4. EEA1 is not a transmembrane protein and should not be drawn as if it were.
- Part 2.3. Most of the references are mostly reviews published before 2005. More recent reviews dealing with glutamate receptor trafficking in synaptic plasticity, should be cited. Given its central role, key results showing AMPAR constitutive cycling should be explicated, like the infusion of tetanus toxin or the antibody feeding experiments, and the primary articles cited.
- The exocyst complex (part 3). The structure of the complex has been indeed deciphered by Picco et al. 2018, but also solved with cryoEM (Mei et al. Nature Struct 2018). This publication must be cited and could be at the basis of the structure presented in Figure 3. In this Figure, the interactions must be made much clearer with appropriate color coding. The documented interactions with PM/Rabs/RalA/TC10 should be indicated. The type of mutant (dominant negative) used in various studies and its effect on complex stability or binding to proteins should be explicated.
- Part 4-6. The first part on the role of exocyst in neuronal development is new and well written but the second part on receptor trafficking is poorly structured. Subdivision in chapters on development/receptor trafficking/synaptic transmission would help. Reference 145 (Gerges et al. 2006) is indeed central to this review, but its discussion is spread over parts 4, 5 and 6. It should be rewritten. In Part 6, we can read at last the central model of AMPAR trafficking and plasticity (see my point 5), but other parts are merely repetitions (e.g. lines 356-9)
Line 420-1. Rephrase “has been observed in recycling conditions…”
Use the current nomenclature for AMPAR and NMDAR subunits (GluA1-4, GluN1, N2A-D). This was the case page 4 but not page 11 and up.
Lines 436-7: What do you mean by “…to participate to LTP processes individually or synergistically” NMDAR activation is necessary for induction, but the expression of LTP is largely mediated by AMPAR trafficking.
Author Response
- Comment: In this manuscript entitled « glutamatergic receptor trafficking and delivery: role of the exocyst complex », Lira et al. review a large body of literature on several subjects dealing with neuronal development (neurite outgrowth and synaptogenesis), synaptic transmission and post-synaptic receptor trafficking in neurons. In addition, the authors review the structure of the exocyst, a complex of 8 proteins conserved from yeast to mammals, and its role in various cellular contexts, mainly in the exocytosis of post-Golgi transport vesicles. In the last part, the authors review the involvement of the exocyst complex in post-synaptic glutamate receptor trafficking. Deciphering the role of the exocyst complex, a central player in intracellular trafficking and exocytosis, in the membrane trafficking of post-synaptic receptors, is an important question in current cellular neuroscience. Therefore, the topic of this review should be of great interest to the readership of Cells. However, I find many parts of the review not clear, some information is repeated while other important data is missing. In conclusion, this review needs to undergo major changes before it is suitable for publication.
Answer: We appreciate that the reviewer considered the subject of this review as something of interest to the readers of the journal. We hope that the corrections we make in this new version are appropriate according to the comments made.
- Comment: Abstract: only post-Golgi trafficking is evoked but for AMPAR trafficking this pathway is not central. Recycling endosomes (REs) are important for trafficking from the endoplasmic reticulum (ER) and for recycling. This should be mentioned in the Abstract.
Answer: Corrected, This was mentioned in the abstract
- Comment: Line 23: please replace “superior eukaryotes” by multicellular organisms, or vertebrate/mammalian cells (also line 74)
Answer: Corrected
- Comment: Line 26: not “vesicle-containing receptors” but the reverse
Answer: Corrected
- Comment: Introduction: it gives a succinct overview of the architecture of synapses and its function. In mentioning the diversity of AMPARs (line 55), the kinetics of channel opening, closing and desensitization should be evoked as well. The connection with the exocyst complex with membrane trafficking in neurons should be presented already in this paragraph, perhaps by evoking the unique challenges a neuron has to face in terms of receptor trafficking.
Answer: In this section, we incorporate a paragraph (line 60-71 in the new version of the manuscript) with more details on the characteristics of glutamatergic receptors, including AMPARs.
- Comment: Part 2. Line 106-108. It is not true that the exocyst is the only tethering factor identified for exocytosis. For synaptic vesicles, Munc18 and 13 play an essential role (see for example review by Brunger et al. 2019). The role of Rab GTPases could be expanded. The part on SNAREs needs to be rewritten. In particular, Jurado et al. 2013 have documented the role of syntaxin3, SNAP47 and VAMP2 in synaptic plasticity. Also, SNAP23 and 25 are not v SNAREs.
Answer: As part of the response to reviewer 1, we incorporated a better explanation regarding the role of the Rabs and other components (SNAREs and others) of the vesicular traffic machinery. The new version of the manuscript includes a rewritten and expanded SNARE part (line 126-138) and an expanded Rab discussion (line 209-221).
- Comment: Part 2.2. This part is very difficult to read. An expanded scheme from figure 2 with the proteins at play would help. There is no mention of proteins associated with GluA1-4 (TARPs, CNIH, FRRS1L) which are important for AMPAR trafficking.
Answer: We appreciate the reviewer's comment. To respond to these requirements, we made an important modification in figure 2 (Part A), by incorporating additional elements of the AMPARs traffic, including their associated proteins, this was also reflected in the text in this section (line 187-208 in the actualized version of the manuscript).
- Comment: Line 159. In ref 66 the colocalisation with EEA1 is only significant in very immature neurons, before synapse formation. It should not be emphasized here, or in Figure 4. EEA1 is not a transmembrane protein and should not be drawn as if it were.
Answer: We found this comment made by the reviewer useful, and decided to delete the comment regarding the role of EEA1. In figure 4 we make the required modifications to the drawing.
- Part 2.3. Most of the references are mostly reviews published before 2005. More recent reviews dealing with glutamate receptor trafficking in synaptic plasticity, should be cited. Given its central role, key results showing AMPAR constitutive cycling should be
explicated, like the infusion of tetanus toxin or the antibody feeding experiments, and the primary articles cited.
Answer: We agree with the reviewer’s comment. More recent references were added in part 2.3:
Ref. 50. Buonarati et al., 2019. DOI: 10.1126/scisignal.aar688.
Ref. 94. Herring et al., 2016. DOI: 10.1146/annurev-physiol-021014-071753
Ref. 97. Volianskis et al., 2010. DOI: 10.1016/j.brainres.2015.01.016
Ref. 99. Jurado et al., 2013. DOI: 10.1016/j.neuron.2012.11.029
Ref. 100. Bin et al., 2018. DOI: 10.1016/j.celrep.2018.05.026
Ref. 103. Moretto et al., 2018. DOI: 10.3389/fncel.2018.00286
Ref. 104. Chiu et al., 2017. DOI: 10.1016/j.neuron.2017.02.031
Ref. 105. Awasthi et al., 2019. DOI: 10.1126/science.aav1483
- The exocyst complex (part 3). The structure of the complex has been indeed deciphered by Picco et al. 2018, but also solved with cryoEM (Mei et al. Nature Struct 2018). This publication must be cited and could be at the basis of the structure presented in Figure 3. In this Figure, the interactions must be made much clearer with appropriate color coding. The documented interactions with PM/Rabs/RalA/TC10 should be indicated. The type of mutant (dominant negative) used in various studies and its effect on complex stability or binding to proteins should be explicated.
Answer: Following the reviewer's comment, we incorporated in section 3 the indicated reference (Mei et al 2018) in line 303-307, but more importantly, we modified Figure 3 to better explain the components of the complex and their interactions (Figure 3A, and 3B; line 315-323).
- Comment: Part 4-6. The first part on the role of exocyst in neuronal development is new and well written but the second part on receptor trafficking is poorly structured. Subdivision in chapters on development/receptor trafficking/synaptic transmission would help. Reference 145 (Gerges et al. 2006) is indeed central to this review, but its discussion is spread over parts 4, 5 and 6. It should be rewritten. In Part 6, we can read at last the central model of AMPAR trafficking and plasticity (see my point 5), but other parts are merely repetitions (e.g. lines 356-9)
Answer: Although the reviewer's suggestion to reorganize an important part of the text is good, we decided to incorporate the requested changes (receptor trafficking) while maintaining the structure of the review. We removed the elements that could be considered repeated and we think the reading is more fluent.
- Comment: Line 420-1. Rephrase “has been observed in recycling conditions…”
Answer: Corrected
- Use the current nomenclature for AMPAR and NMDAR subunits (GluA1-4, GluN1, N2A-D). This was the case page 4 but not page 11 and up.
Answer: In this version we take care to use the appropriate nomenclature for glutamatergic receptors, making the necessary changes in each mention in the text.
- Comment: Lines 436-7: What do you mean by “…to participate to LTP processes individually or synergistically” NMDAR activation is necessary for induction, but the expression of LTP is largely mediated by AMPAR trafficking.
Answer: To avoid a confusing reading, we removed the mentioned phrase from the text.
Reviewer 3 Report
This review is interesting, well organized, well-written, biologically plausible, and appropriately controlled. It described the ion channel trafficking during synaptic formation and LTP/LTD generation in the neuron and glial cells. The overall review is easily understandable and well discusses the recent trend and results. In particular, the description of the role of the exorcist complex was fresh in an area that was not covered much in previous reviews. it is a well-performed and well-articulated review that I believe, aside from a few minor grammatical errors, is ready for publication and of interest to your readership.
Author Response
Comment: This review is interesting, well organized, well-written, biologically plausible, and appropriately controlled. It described the ion channel trafficking during synaptic formation and LTP/LTD generation in the neuron and glial cells. The overall review is easily understandable and well discusses the recent trend and results. In particular, the description of the role of the exorcist complex was fresh in an area that was not covered much in previous reviews. it is a well-performed and well-articulated review that I believe, aside from a few minor grammatical errors, is ready for publication and of interest to your readership.
Answer: We appreciate all the positive comments made by the reviewer of our work, we hope that the modifications made will remain in line with what the reviewer indicated.
Round 2
Reviewer 2 Report
The authors have answered all my queries and increased the quality of the manuscript. The manuscript is now suitable for publications. I still have a couple of remarks:
Abstract: line 22. The exocyst is not just about Golgi-derived vesicles. REs probably contribute (see Gerges et al.). Please remove “Golgi-to-plasma membrane”
Introduction: line 60. AMPARs have 4 agonist sites, not 2. Please change. Same for NMDARs (64). You could also mention that D-serine probably substitutes for glycine in activating NMDARs in physiological contexts (see e.g. Papouin et al. Cell 2012)
Author Response
- Comment: The authors have answered all my queries and increased the quality of the manuscript. The manuscript is now suitable for publications. I still have a couple of remarks:
Answer: We appreciate the reviewer's comment, and we hope that with these latest improvements the manuscript will move to the next stage in the publication process.
- Comment: Abstract: line 22. The exocyst is not just about Golgi-derived vesicles. REs probably contribute (see Gerges et al.). Please remove “Golgi-to-plasma membrane”
Answer: The indicated phrase was removed from the abstract
- Comment: Introduction: line 60. AMPARs have 4 agonist sites, not 2. Please change.
Answer: this was modified
- Same for NMDARs (64).
Answer: this was modified
- You could also mention that D-serine probably substitutes for glycine in activating NMDARs in physiological contexts (see e.g. Papouin et al. Cell 2012)
Answer: According to the reviewer's suggestion this was modified